# Impact of Oat Supplementation on the Structure, Digestibility, and Sensory Properties of Extruded Instant Rice

**DOI:** 10.3390/foods13020217

**Published:** 2024-01-10

**Authors:** Junling Wu, Kai Zhu, Sijie Zhang, Meng Shi, Luyan Liao

**Affiliations:** College of Food Science and Technology, Hunan Agricultural University, Changsha 410128, China; wujunling1114@163.com (J.W.); zhukai2000hnau@163.com (K.Z.); zsj18627329702@outlook.com (S.Z.); shimeng@hunau.edu.cn (M.S.)

**Keywords:** instant-extruded rice, oat powder, high dietary fiber, extrusion technology

## Abstract

The addition of oat at varying percentages (26%, 32%, 38%, 44% and 50%) was used to evaluate the structural, microstructural, and physicochemical changes in instant-extruded rice (IER). A mixture of broken rice and oat flour was extruded in a twin-screw extruder. It was found that when adding 44% oats, the gelatinization degree of the mixed powder was the lowest (89.086 ± 1.966%). The dietary fiber content increased correspondingly with the increase in oat addition. Analyses of texture properties revealed that the hardness, adhesive, and resilience values increased and then decreased with oat addition. Compared with other common instant rice (IR), the advantages of IER were evaluated in terms of microstructure, digestive performance, and flavor. IER with 44% oat addition obtained in this study had higher hardness, adhesiveness, rehydration time, and sensory score, and the content of resistant starch (RS) reached 6.06%. The electronic nose and electronic tongue analyses could distinguish the flavor of different IR efficiently. This study showed the feasibility of preparing fiber-enriched IER. The results demonstrated the potential for the development and utilization of broken rice, providing a reference for the development of IER.

## 1. Introduction

With the development of society and economy, the pace of life is increasing, and the demand for instant food is also growing. In the face of COVID-19, the need for instant foods is even more urgent. Instant rice not only conforms to the staple food habits of the public, but also meets the demand for convenience and nutrition of instant food, and has a promising market [1]. However, most traditional instant rice products exhibit poor eating quality and low consumer acceptance due to aging problems, while extruded rice is a good solution to this problem [2]. Compared to the traditional method of processing rice, extrusion can improve the processing of instant rice and increase the added value of the ingredients [3,4]. Instant-extruded rice (IER) has begun to show a trend toward incorporation of multigrain and enhanced nutrition, and the production of high-nutrition IER has become the main research direction of many scholars [5,6]. During the production process of rice, a lot of broken rice is produced as a by-product of the rice after shelling and polishing [4]. How to effectively utilize low-value broken rice is a key issue. Broken rice has been documented to contain several bioactive compounds [7], and demands a lower price than rice and corn, so it has a certain application prospect [8]. Extrusion processing could modify the microstructure and digestibility of broken rice effectively [9], which is widely used in starch-containing food products [10].

Oats are rich in dietary fiber and active ingredients, which can be used as a supplement to make up for the lack of dietary fiber intake and provide various health benefits including hypoglycemic, hypolipidemic, immune-boosting, antioxidant, and anti-ageing properties [11,12,13]. However, the high fiber content and poor shear resistance of the starch granules in oats can lead to issues with transparency, texture, and taste that limit food applications [14]. Pretreatments and processing conditions play an important role in determining oat ingredient quality, as factors such as steaming have been found to increase the development of volatile flavor compounds [15,16,17], while sprouting elicits changes in digestibility and nutrition [18]. Modifications like heat–moisture treatment have also shown promise in improving oat starch properties like thermal stability and retrogradation [19]. Recently, extrusion technology is becoming increasingly sophisticated in the food industry, primarily applied to ready-to-eat food products, usually rice and oats [20]. Extrusion processing has emerged as an effective means of enhancing the hydration, solubility, and viscosity of oat flours and other components [21].

Despite the health benefits of oats, there are few reports on the comprehensive effects of oat addition on the structure, digestion, and flavor characteristics of IER products. Therefore, steamed oats and crushed broken rice were used as raw materials to produce IER through extrusion processing to determine these characteristics. According to the comprehensive evaluation index for the quality of IER established in our previous research, IER with the highest comprehensive scores was selected for subsequent comparison [22]. At the same time, comparative analysis was conducted from multiple perspectives with extruded rice samples including instant-extruded rice with optimal oat addition (IER-OA), instant-extruded rice with raw oats (IER-RO), commercially available oat instant rice (CAIR) and commercially available extruded instant rice (CAER). Therefore, the aim of this study was to develop a high-dietary-fiber IER and evaluate the effect of oat addition on the structure, digestion, and flavor quality of IER.

## 2. Materials and Methods

### 2.1. Materials

Broken rice was purchased from Jiao Shan Rice Co., Ltd. (Changsha, China). Oats were purchased from Zhangjiakou Wanquan District Oat Pulse Food Co., Ltd. (Changsha, China). CAIR was purchased by Hubei Shiyuan Food Co., Ltd. (Anlu, China). It contained 3.8% dietary fiber according to the product specifications. CAER is from Shanmi Technology Co., Ltd. (Chengdu, China).

### 2.2. Preparation of Oat–Rice Flour Blends

The oat seeds were spread flat in a steamer drawer, placed in an electric steamer and steamed in water for 25 min, then placed in a constant temperature drying oven at 40 °C for 24 h. Oats and broken rice were crushed and passed through 80 meshes sieve and stored in a sealed container at 4 °C, respectively. The content of oat flour in broken rice flour was in the proportions of 26%, 32%, 38%, 44%, and 50%, respectively.

### 2.3. Pasting Characteristics of Oat–Rice Flour Blends

The pasting properties of oat–rice flour blends were analyzed by a rapid viscosity analyzer (RVA-TecMaster, Perten Instruments, Stockholm, Sweden). The pasting indexes, including peak viscosity (PV), trough viscosity (TV), final viscosity (FV), pasting temperature (PT), breakdown (BD) and the setback value (SV), were recorded.

### 2.4. The Extrusion Process

The oat–rice flour blends were processed in a twin-screw extruder (FWHE36-24, Fumarco Food Engineering & Technology Co., Changsha, China), with five constant heating temperatures (65 °C, 110 °C, 140 °C, 90 °C and 80 °C). The solid feed rate was 12 kg/h, and the water feed rate was 19%. The screw speed was 190 r/min. After extrusion, the cutting machine was equipped with four blades and a speed of 2000 rpm. After cutting, the extruded rice grains were sent into the fluidized bed (30 °C) (FMFC, Fumarco Food Engineering & Technology Co., Changsha, China) with the feeder for preliminary drying for 2 min, and then sent into the continuous dryer (the temperature of the first zone was 130 °C, and the temperature of the second zone was 140 °C) (FMDR, Fumarco Food Engineering & Technology Co., Changsha, China) for drying for 3 min, and then taken out and cooled flatly. IER was prepared for testing.

### 2.5. The Properties of IER

#### 2.5.1. Hydration Properties

A 1.0 g sample was dispersed in 40 mL distilled water and stirred for 30 min at 35 °C in a thermostat water bath (DK-98-II, Taist Instrument Co., Ltd., Tianjin, China). The suspension was centrifuged (TDZ5, Hunan Xiangyi Laboratory Instrument Development Co., Changsha, China) at 4500 r/min for 30 min. The supernatant was dried in a hot air oven at 105 °C to a drying oven (GFL-230, Lebo Terui Instrument Co., Ltd., Tianjin, China) at 105 °C to a constant weight. The water absorption index (WAI), the swelling power index (SPI), and the water solubility index (WSI) were calculated according to the following equations:(1)$$\mathrm{WSI}\left(\%\right)=\frac{\left(M1-M0\right)}{m}\times 100$$
(2)$$\mathrm{WAI}(g/g)=\frac{\left(M2-M\right)}{m}$$
(3)$$\mathrm{SP}(g/g)=\frac{\left(M2-M\right)}{m(100-\mathrm{WSI})}$$
where *m* is the sample mass (g), *M* is the centrifuge tube mass (g), *M*0 is the constant weight bottle dish mass (g), *M*1 is the constant weight supernatant and flat dish total mass (g), and *M*2 is the precipitate and centrifuge tube total mass (g).

#### 2.5.2. Dietary Fiber Content

As described elsewhere [23], the dried IER were digested by α-amylase, protease, and glucosidase to remove protein and starch, then precipitated by ethanol, filtered, and the residue was washed with ethanol and acetone, and then dried and weighed to obtain the total dietary fiber residue m_R_, the mass m_a_ and m_p_ of ash and protein in the residue were measured separately, and m_b_ was obtained through blank experiments. The formula for dietary fiber content is as follows:(4)$$\mathrm{Dietary}\;\mathrm{fiber}\;\mathrm{content}\left(\%\right)=(m_{R}-m_{p}-m_{a}-m_{b})/m$$
where m is the sample mass (g), m_R_ is the residue mass (g), m_a_ is the ash mass (g), m_p_ is the protein mass (g), and m_b_ is the blank mass (g).

#### 2.5.3. Rehydration Characteristics

IER (100 g) was mixed with 5-fold boiling water in a beaker, incubated in a water bath for 5 min, then the IER was placed on a glass plate and covered with the same glass plate until there was no white inner core. The final time spent was the rehydration time for IER. Rehydration rate: Accurately weighed IER as *M*1(g), added at 5 fold the mass of boiling water, which was kept until the rehydration time. We dried the surface of the rice grains with absorbent paper, which were weighed as *M*2 (g), and the rehydration rate (%) was expressed as *M*2/*M*1.

#### 2.5.4. Texture Profile Analysis (TPA)

IER were heated by a rice cooker at 100 °C for 30 min. The sequence of procedures was as follows: heated and cooked for 25 min, and reduced to room temperature for 30 min. Then, the sample container was separated into three layers, with a total of 8 grains of rice taken from each layer, which were evenly and symmetrically placed on the carrier table for testing. The texture profile of IER was measured by a texture analyzer (TA-XT2i Plus, Stable Micro System Co., Ltd., Surrey, UK) with a P 36/R probe, which compressed rice grains up to 50% with a trigger force of 5 g. The test speed was 0.5 mm/s. The final measurement result was the average value after removing the maximum and minimum values in each of the three parallels (10 times).

#### 2.5.5. Sensory Evaluation

Sensory evaluation scoring rules for IER were developed to evaluate the odor, appearance structure (color and form), palatability (stickiness, chewiness, softness, and hardness), taste and texture. An evaluation team consisting of 15 food professionals was formed to score the sensory quality of the rehydrated and the naturally cooled IER one by one according to the rules of sensory evaluation. Each sample was repeated three times to eliminate errors and calculate the average value. The evaluators had a thorough understanding of the situation of this study before conducting the research, and the IER samples met the food safety standards of Hunan JiaoShan Rice Industry. The personal privacy rights of the sensory personnel were fully protected.

### 2.6. Comparison of Four Kinds of Instant Rice (IR)

#### 2.6.1. Microstructure

The microscopic observations of IER were observed by scanning electron microscopy (SEM, S-3000N, HITACHI, Tokyo, Japan). Each sample was secured to a load table with conductive double-sided tape and then sprayed with gold under vacuum conditions. Then, the samples were observed and photographed under a scanning electron microscope with a magnification of 300 and 5000, respectively.

#### 2.6.2. In Vitro Digestion

The freeze-dried samples were crushed and sieved through 100 mesh. To an accurately weighed 10 mg of sample, we added 8 mL of sodium acetate buffer (pH = 5.2, 3 mol/L), and shook the mixture well. The samples were heated in a boiling water bath for 20 min and then insulated at 37 °C. Then, 1 mL of 37 °C pepsin (2500 U/mL) solution was added to the sample and it was shaken at 37 °C for 20 min (120 r/min), and then 1.0 mL of enzyme mixture (170 U/mL α-amylase and 21 U/mL amyloglucosidase) was added and shaken in water at 37 °C for 150 min (120 r/min). At 0 min, 20 min and 120 min, respectively, 1.0 mL of hydrolysis solution was taken into a 25 mL graduated cuvette, 4 mL of sodium hydroxide solution (0.4 mol/L) was added to inactivate the enzyme, and then 2 mL of DNS reagent was added, and this mixture was heated in a boiling water bath for 5 min. After cooling, it was brought to 25 mL with distilled water, and the absorbance was measured at 540 nm. A volume of 1 mL of distilled water was taken as a blank control. The equation of the measured glucose standard curve was y = 0.4006x − 0.0009, R^2^ = 0.9975, and the glucose content in the supernatant of the sample was calculated and multiplied by 0.9 to finally obtain the content of rapidly digestible starch (RDS), slowly digestible starch (SDS) and resistant starch (RS) within the sample. The formulas were as follows:(5)$$\mathrm{RDS}/\%=\frac{G20-FG}{\mathrm{TS}}\times 0.9$$
(6)$$\mathrm{SDS}/\%=\frac{G120-G20}{\mathrm{TS}}\times 0.9$$
(7)$$\mathrm{RS}/\%=\frac{\mathrm{TS}-\mathrm{RDS}-\mathrm{SDS}}{\mathrm{TS}}$$
where *G*20 is the glucose content in 20 min (mg), *FG* is the glucose content before hydrolysis (mg), *G*120 is the glucose content in 120 min (mg), and TS is the total starch content (mg).

#### 2.6.3. The Electronic Nose

The flavor profiles of IR were analyzed by an electronic nose (PEN3, AIRSENSE Analytics GmbH, Schwerin, Germany), consisting of 10 metal oxide sensors to differentiate the different types of flavor substances. The types of sensors and their corresponding sensitive substances are shown in Table 1. Samples were blended in boiling water at a material–liquid ratio of 1:1.2 and rehydrated for 8 min, respectively. Samples (3 ± 0.05 g) were placed in a 20 mL headspace flask and incubated in a water bath at 70 °C for 30 min. The measurement conditions were listed as follows: the sampling time was 1 s/group, the sensor self-cleaning time was 120 s, the sensor zeroing time was 10 s, the sample preparation time was 5 s, the injection flow rate was 400 mL/min, and the analysis sampling time was 120 s. The results are summarized by principal component analysis (PCA). The flavor substances were determined by direct headspace aspiration method using different sensors, and triplicate sets were made for each sample.

#### 2.6.4. The Electronic Tongue

The taste property of IR was assessed by an electronic tongue (TS-5000Z, INSTENT, Fukuoka, Japan), comprising 9 sensors. The cooked sample (15 g) was mixed with 150 mL pure water, homogenized for 1 min, centrifuged at 7000 rpm for 3 min, and the supernatant was taken for electronic tongue analysis. Three parallel sets were made for each sample.

### 2.7. Statistical Analysis

The results were expressed as the mean ± standard error. SPSS Statistics 25.0 software was used for descriptive analysis and significance analysis of the experimental data. The figures were plotted using Origin 2021 software.

## 3. Results and Discussion

### 3.1. The Properties of Oat–Rice Flour Blends

The addition of oat to broken rice significantly affected (*p* < 0.05) the pasting indexes (Table 2). Peak viscosity (PV) was the highest degree of starch gelatinization occurs, which was related to the volume occupied by the swelling capacity of the starch [24]. Trough viscosity (TV) was caused by the degradation of starch at high temperatures, resulting in a decrease from the peak value in peak viscosity. TV dropped first and then rose with increasing oat addition. Breakdown (BD) first increased from 26% to 44% and then decreased at 50%, and the difference between all samples was significant (*p* < 0.05); the increase in BD decomposition value indicated that the shear force could easily break down the fully swollen starch granules [25]. The low pasting temperature (PT) of oat–rice flour blends was caused by the lower amount of starch that needed to be gelatinized in the highly gelatinized oat–rice flour blends and the damaged starch granule was more accessible to gelatinize [26]. The setback value (SV) represents the retrogradation trend of cooled oat starch, which was related to the presence of amylopectin fragments and the amylose leaching content [27]. The sample with 26% oat addition had the lowest breakdown (355.00 ± 45.90 Pa·s) and setback values (2382.33 ± 51.00 Pa·s) with a more stable network and higher anti-aging ability of oat–rice flour blends [28]. Steaming as a pretreatment of oats impacts viscosity by inducing changes in the molecular structure of oat starch, destruction of the molecular chain structure, the formation of a starch–lipid complex and other reasons that can lead to changes in viscosity of oat under high temperature and high moisture conditions [29]. SV and PT gradually climbed with the increase in oat addition. Another factor influencing the pasting properties of oat–rice flour blends is the presence of dietary fiber, due to its replacement of water-swelling starch [30].

### 3.2. IER with Different Amounts of Oat Added

#### 3.2.1. Gelatinization and Hydration Characteristics

Both before and after extrusion, the gelatinization degree (GD) showed a similar trend of first decreasing and then increasing with greater oat addition (Figure 1A). But the difference in GD before extrusion was more significant, with GD being the smallest when adding 44% oats. After reaching the gelatinization temperature, IER was gelatinized, and the structure was disintegrated, and GD increased during heating [31]. The water solubility index (WSI) (Figure 1B), the water absorption index (WAI) (Figure 1C) and the swelling power index (SPI) (Figure 1D) of IER after extrusion were higher than those before extrusion. Due to the physical effects of extrusion, the starch particle structure changes, and this change made the IER have stronger water absorption and led to an increase in soluble nutrients, which was conducive to improving the nutritional value of the product [32,33]. The WSI before and after extrusion also gradually increased with oat addition, suggesting that the soluble substances in it also increased, potentially due to the high proportion of damaged starch granules in oat flour, which allowed more water into oat flour particles [34]. With an increase in the content of oat supplementation, more composites, and an increase in fiber content in the material were present, resulting in a decrease in WAI and SPI.

#### 3.2.2. Dietary Fiber Content

With the increase in oat addition, the dietary fiber content of IER increased, and the difference between samples was significant (*p* < 0.05) (Figure 2A). The dietary fiber content of IER exceeded the requirement standards of high-dietary-fiber food, products were considered high in dietary fiber when they contained >6% [6], and the dietary fiber content of IER with 50% added amount reached 13.24%, more than 2 fold that of the minimum specification requirement of high-dietary-fiber food. Extrusion was an important method to increase the content of soluble dietary fiber [35].

#### 3.2.3. Rehydration Characteristics

Rehydration is mainly to enable the rice to fully absorb water and create necessary conditions for starch gelatinization. The rehydration time of IER increased with the increase in oat addition, and the rehydration rate decreased from 26% to 38%, and then increased from 38% to 50% (Figure 2B). Complete soaking allows the water to fully penetrate the rice grain, allowing the starch to gelatinize quickly and fully. If the rehydration time was too long, the soluble nutrients in rice would be lost, and the whiteness of rice would also decrease, seriously affecting the quality of IER. If the rehydration time was too short, it would be easy to make IR crispy and taste worse. Therefore, controlling the appropriate rehydration conditions is very important for the taste and nutritional preservation of the final food products.

#### 3.2.4. Sensory Score

The effect of oat addition on the sensory evaluation of IER is shown in Figure 2C. The sensory score showed a trend of first increasing and then decreasing, reaching the highest score of 81.25 with 38% oat addition. This indicated that appropriate addition of steamed oats could help improve the sensory quality of IER. Although dietary fiber has good nutritional function, it is still necessary to comprehensively consider the quality of the product and the demand for dietary fiber when designing high-fiber extruded rice products.

#### 3.2.5. Textural Measurement

As shown in Table 3, the CV of chewiness, hardness and adhesiveness was higher, indicating that the level of oat addition had a greater effect on these parameters. With the increase in oat addition, the hardness and adhesiveness increased and then decreased, reaching peak values at 44% oat addition; the cohesiveness decreased; the resilience increased first and then decreased; the springiness and chewiness had no significant changes (*p* > 0.05), which might be related to the starch property and nutritional composition of oat. The content and proportion of amylose and amylopectin in oat starch also affected the gel properties of starch, thus altering the cohesion and resilience of IER [36].

### 3.3. Comparison of Four Kinds of IR

#### 3.3.1. Texture, Rehydration Time, and Sensory Score

The hardness and adhesiveness were the highest among the CAIR texture characteristic, and the sensory score was the highest (Figure 3), the loss of moisture in CAIR produced firming and crust formation, but also the recrystallization of starch would contribute to firmness [37]. After the second high temperature rehydration, the dissolution of soluble substances in the rice grains was accelerated, which reduced the hardness and adhesiveness of other IR. CAER had higher hardness, adhesiveness, and rehydration time, which might be related to the food additives added during the rice preparation process; Compared to IER-RO, IER-OA had higher hardness, adhesiveness, rehydration time, and sensory score, its quality was closest to that of CAER, mainly due to the addition of steamed oat flour which exhibited higher starch hydrolysis [28]. The changes in these indicators have a certain impact on the edible quality of IER.

#### 3.3.2. Microstructure

SEM results of four types of IR (Figure 4) revealed the kernel microstructure, which depended on the arrangement of molecules inside the granules. The imaging of the four samples at a magnification of 300 times (Figure 4A1–A4) showed the surface of the three IER granules was more rough and irregularly shaped, whereas the starch granules of CAIR were larger, with most of the surfaces being smoother and with better granule integrity. As the results of previous studies, under high-temperature and high-pressure conditions of extrusion, oat-broken rice starch was heated and expanded, and part of amylopectin was decomposed into small molecules such as maltodextrin and amylose, resulting in increased surface fragments and improved surface roughness of starch particles [38,39]. IER-OA had small and numerous surface fragments. Although the surface of starch granules was also relatively rough from IER-RO, the particle surface fragments were larger than those of IER-OA, and the reason for this might be related to the addition of raw oats as well as the amount of oats added to IER, which made the internal structure more loosely packed. After magnifying 5000 times (Figure 4B1–B4), CAIR had fewer internal pores and smaller pore sizes, with a dense internal structure, whereas the other three samples all had different sizes and dense pores, which indicated that the action of the extruder caused the starch to break down, which made the internal structure of the samples looser than that of the fresh rice samples. Among them, IER-RO had the largest number of pores with different pore sizes. Compared with IER-RO and CAER, IER-OA had fewer internal pores and smaller pore sizes. During the extrusion process, both carbohydrate and protein structures underwent changes, which affected the expansion and density of the extruded product [40].

#### 3.3.3. In Vitro Digestion

The proportion of the rapidly digestible starch (RDS), slowly digestible starch (SDS) and resistant starch (RS) of four IR are shown in Figure 5. Depending on the rate of glucose release and absorption in the gastrointestinal tract, where SDS leads to a slower entry of glucose into the bloodstream, a lower glycemic response and complete digestion in the small intestine at a lower rate than RDS, whereas RS is a starchy fraction that is indigestible in both the stomach and small intestine and functions similarly to dietary fiber, making RS beneficial for promoting intestinal health and lowering blood glucose levels.

From Figure 5, the RS content was the least in CARE without oat addition, while RDS content was the highest. Compared with CAER, IER-RO showed a significant decrease in RDS content (*p* < 0.05), a significant increase in SDS content (*p* < 0.05), and an increase in RS content, with the highest SDS content of 49.38%. On the one hand, the addition of oats increased the content of dietary fiber in IR. On the other hand, the lipid rich in oats formed a starch–lipid complex during the extrusion process [33], and the increase in substances inhibited the digestion of IER-RO in vitro. Among the three kinds of IR with oats, RS content in CAIR was the highest, reaching 7.05%, and RDS content was also the highest, mainly because the dietary fiber content in CAIR was the highest (up to 23%), and RDS content was relatively high. The content of RS of IER-OA reached 6.06%, which was similar to that of CAIR, and the SDS content was more than CAIR, and the RDS content was less than CAIR. The main reason was that extrusion processing, therefore, significantly reduced the digestibility of starch, by significantly increasing the SDS and RS contents [41]. The starch macromolecules degraded to a certain extent after extrusion, the hydrogen bonds between starch molecules have broken, causing changes in the structure of starch, resulting in some starch gelatinization and decomposition into low-molecular oligosaccharides, promoting the digestion and absorption of starch [42]. In summary, its anti-digestive ability was higher than CAIR. From the perspective of nutrition, IER-OA had the best comprehensive quality.

#### 3.3.4. The Electronic Nose

Principal component analysis (PCA) was used to reduce the complexity of analyzing the similarities and differences among the data of different samples by dimensional reduction. The PCA results of the electronic nose for samples were shown in Figure 6A—the contribution rate of PC1 was 69.1%, the contribution rate of PC2 was 30.7%, and the total cumulative distinguishing contribution rate of 99.9%, which covers most of the flavor information. IER-OA and IER-RO were similar and partially overlapped, indicating similar flavor characteristics. CAIR was far away from that of the others, showing that it differed from the other samples in terms of flavor.

As shown in Figure 6B, the response intensity values of the W1S, W2W, W1W, W2S and W5S sensors were powerful in four IR, signifying higher concentrations of short-chain alkanes and organic sulfides. Regarding the overall odor profile, IER-OA and IER-RO were similar. Compared to the other two samples, the response values of the W1W and W2W sensors were higher at IER-OA and IER-RO, and CAIR had a slightly higher response on the W1S sensor than the other samples. CAER was significantly weaker than the other samples in overall sensor response values. Thus, the electronic nose analysis could distinguish the flavor of different IR efficiently.

#### 3.3.5. The Electronic Tongue

The PCA results of the electronic tongue of the four IR were shown in Figure 6C. The variance contributions of PC1 and PC2 of the IR were 73.1% and 22.4%, respectively, and the cumulative variance contribution reached 95.5%, which reflected the overall taste information of the samples effectively. The PC1 was mainly the bitterness and the PC2 was the sweetness and umami. CAIR and CAER in the second quadrant were negatively correlated with PC1 and positively correlated with PC2, while IER-RO, IER-OA in the 4th quadrant were negatively correlated with PC1 and negatively correlated with PC2. Additionally, there was no overlap area between samples. The CAER without oats is located farthest and has a different taste compared to the other three IR.

The radar plot of taste value was made from the electronic tongue data in Figure 6B. It can be intuitively seen that the bitterness of the four samples was the most obvious, with certain umami and sweet taste. Astringency, bitter, and umami (richness) were almost to the tasteless point, while sour, astringency, and saltiness were much lower than the tasteless point. Therefore, the effective evaluation indexes of four kinds of IR were bitterness, umami, and sweetness. Among the four samples, the bitterness of CAER was significantly higher than that of the other three samples in terms of bitterness, umami, and sweetness. The bitterness ranking was as follows: CAER > IER-OA > IER-RO > CAIR, and might be related to the storage period and preparation technology of IR. The umami ranking was as follows: IER-OA > IER-RO > CAIR. However, CAER did not have umami, which was mainly related to the addition of oats would increase the umami, might be the oats taste in multiple high temperature formed after the product of the Merad reaction [43]. The sweetness ranking was as follows: CAER > CAIR > IER-RO, the IER-OA had no sweet taste, which might be due to the higher content of oat flour in the IER-OA. Fatty acids produced by the enzymatic hydrolysis of the lipid substances contained in it would produce a certain bitterness, resulting in the reduction in sweetness [44]. Therefore, the electronic tongue analysis could play a significant role in distinguishing the taste properties of the different IR.

## 4. Conclusions

The different oat additions affected the dietary fiber content, pasting and textural characteristics of IER. With the increase in the supplemented oat amount, the hardness and adhesiveness first increased and then decreased, peaking at 44%. The rehydration time gradually increased, and the sensory score showed a trend of first increasing and then decreasing. IER with 44% steamed oats had the highest umami taste and the best comprehensive quality. In addition, IER-OA was compared with IER-RA, CAIR and CAER to investigate the differences in texture, microstructure, flavor, sensory and in vitro digestive characteristics of the four types of IR, to achieve a more objective evaluation of high-dietary-fiber IER. This study has provided a basis to produce high-nutrition extruded food, and provided a new way for the comprehensive development and utilization of broken rice.

## Figures and Tables

**Figure 1 foods-13-00217-f001:**
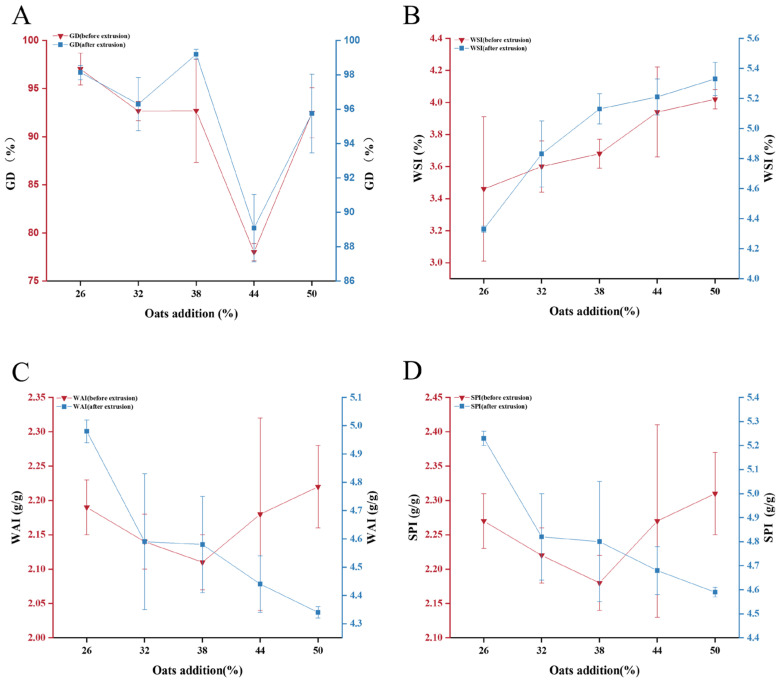
Gelatinization and hydration characteristics of IER with different oat additions. GD, gelatinization degree (**A**); WSI, the water solubility index (**B**); WAI, the water absorption index (**C**); SPI, the swelling power index (**D**).

**Figure 2 foods-13-00217-f002:**
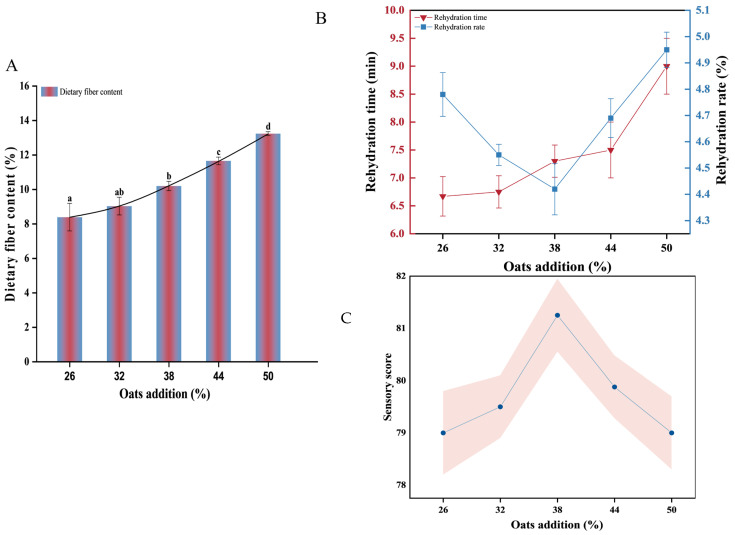
Effect of oat addition on of IER. Dietary fiber (**A**), rehydration time, rehydration rate (**B**), and sensory score (**C**). a–d: different lowercase letters indicated significant differences (*p* < 0.05).

**Figure 3 foods-13-00217-f003:**
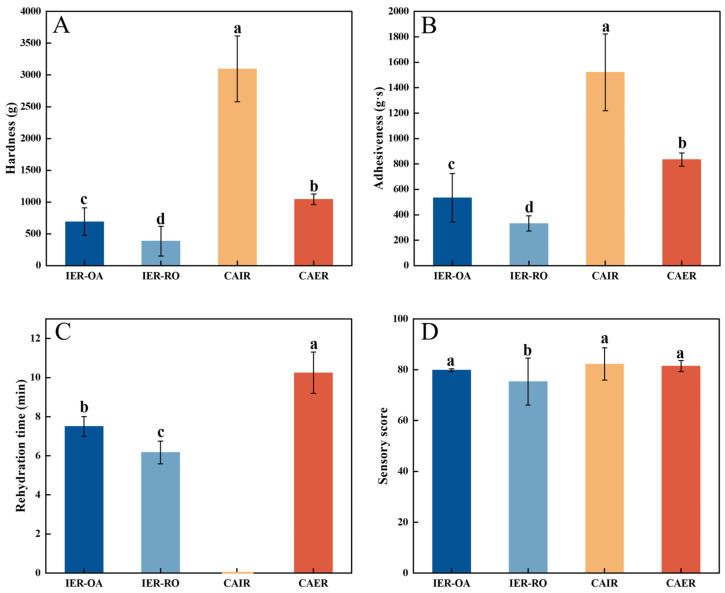
Texture, rehydration time, and sensory score of different IR. Hardness (**A**), adhesiveness (**B**), rehydration time (**C**) and sensory score (**D**). Instant-extruded rice with optimal oat addition (IER-OA), instant-extruded rice with raw oats (IER-RO), commercially available instant rice (CAIR), and commercially available extruded instant rice (CAER). a–d: different lowercase letters indicated significant differences between samples (*p* < 0.05).

**Figure 4 foods-13-00217-f004:**
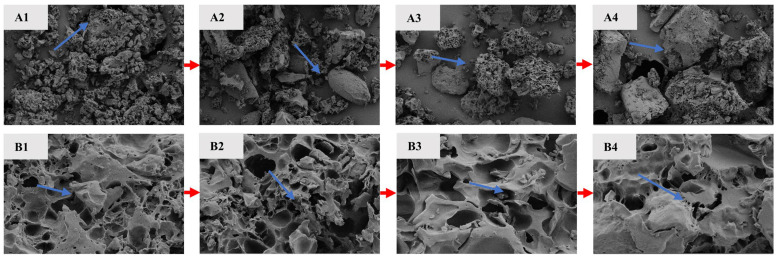
Micromorphology of different IR by scanning electron microscopy. (**A1**–**A4**) is imaged at 300 times magnification; (**B1**–**B4**) is imaged at 5000 times magnification. 1 to 4 is in the order of IER-OA, IER-RO, CAIR, and CAER.

**Figure 5 foods-13-00217-f005:**
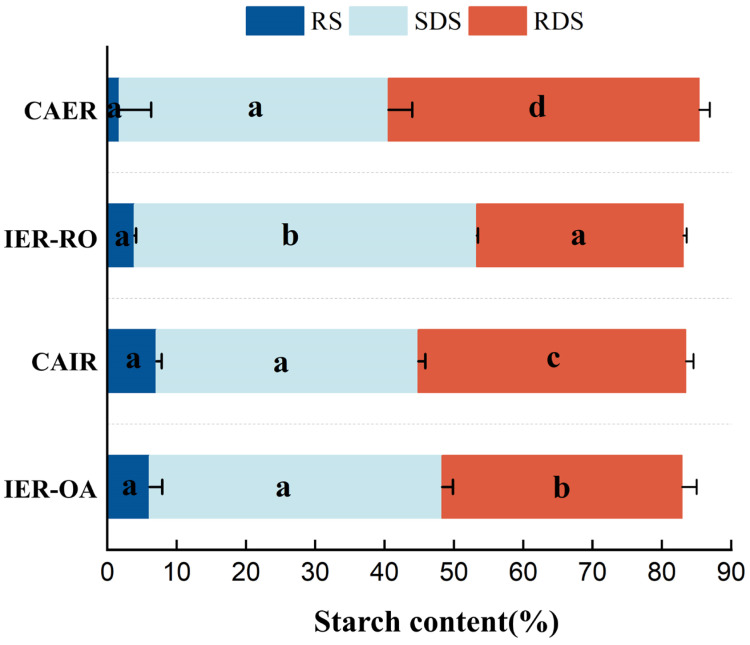
The starch digestibility properties of different IR. Rapidly digestible starch (RDS); slowly digestible starch (SDS); resistant starch (RS). Instant-extruded rice with optimal oat addition (IER-OA), instant-extruded rice with raw oats (IER-RO), commercially available oat instant rice (CAIR) and commercially available extruded instant rice (CAER). a–d: different lowercase letters indicated significant differences between samples (*p* < 0.05).

**Figure 6 foods-13-00217-f006:**
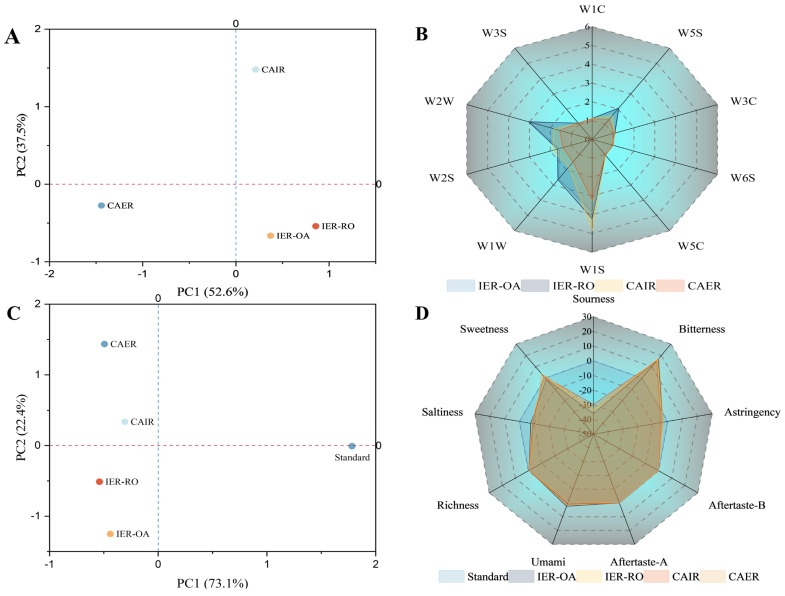
(**A**) PCA diagram of the electronic nose value with different IR, (**B**) radar diagram of the electronic nose value with different IR, (**C**) PCA diagram of the electronic tongue value with different IR, and (**D**) radar diagram of the electronic tongue value with different IR. IER-OA, instant-extruded rice with optimal oat addition; IER-RO, instant-extruded rice with raw oats; CAIR, commercially available oat instant rice; CAER; commercially available extruded instant rice.

**Table 1 foods-13-00217-t001:** Ten sensors and their descriptions in the electronic nose.

No.	Sensors	Descriptions
MOS1	W1C	Aromatic compound
MOS2	W5S	Nitrogen oxide
MOS3	W3C	Ammonia, aromatic component
MOS4	W6S	Hydrogen
MOS5	W5C	Alkanes, aromatic component
MOS6	W1S	Exercise alkanes
MOS7	W1W	Sulfide
MOS8	W2S	Ethanol
MOS9	W2W	Organic sulfides, aromatic components
MOS10	W3S	Alkanes

**Table 2 foods-13-00217-t002:** Effect of oat addition on the gelatinization properties of IER.

Oat Addition	PV (Pa·s)	TV (Pa·s)	BD (Pa·s)	FV (Pa·s)	SV (Pa·s)	PT (°C)
26%	2389.67 ± 54.88 ^bc^	2034.67 ± 100.76 ^c^	355.00 ± 45.90 ^a^	4417.00 ± 62.22 ^ab^	2382.33 ± 51.00 ^a^	85.07 ± 0.06 ^a^
32%	2448.67 ± 24.54 ^c^	1992.67 ± 51.43 ^c^	456.00 ± 37.16 ^b^	4434.00 ± 43.27 ^ab^	2441.33 ± 10.07 ^a^	85.62 ± 0.45 ^b^
38%	2386.00 ± 35.51 ^abc^	1763.33 ± 35.23 ^b^	622.67 ± 65.73 ^c^	4436.33 ± 56.50 ^ab^	2673.00 ± 82.83 ^b^	85.92 ± 0.03 ^b^
44%	2379.67 ± 33.50 ^ab^	1528.67 ± 39.37 ^a^	851.00 ± 53.39 ^e^	4321.33 ± 84.09 ^a^	2792.67 ± 80.77 ^b^	86.50 ± 0.48 ^c^
50%	2319.67 ± 13.01 ^a^	1586.67 ± 16.04 ^a^	733.00 ± 3.46 ^d^	4513.33 ± 73.08 ^b^	2926.67 ± 76.46 ^c^	86.68 ± 0.06 ^c^
MIN	2319.67	1528.67	355.00	4321.33	2382.33	85.07
MAX	2448.67	2034.67	851.00	4513.33	2926.67	86.68
CV/%	1.92	12.89	33.33	1.55	8.72	0.77

Data are the mean ± standard deviation of at least two replicates. Means with different letters within the same row indicate a significant difference at *p* < 0.05. PV: peak viscosity; TV: trough viscosity; BD: breakdown; SV: setback value; FV: final viscosity; PT: pasting temperature; MIN: minimum; MAX: maximum; CV: coefficient of variation.

**Table 3 foods-13-00217-t003:** Effect of oat addition on the texture properties of IER.

Samples	Hardness/g	Springiness/s	Cohesiveness	Adhesiveness/g·s	Chewiness/g	Resilience
26%	511.83 ± 204.10 ^a^	1.02 ± 0.04 ^a^	0.80 ± 0.03 ^b^	410.54 ± 164.21 ^a^	422.03 ± 158.84 ^a^	0.69 ± 0.05 ^a^
32%	532.40 ± 116.30 ^a^	0.97 ± 0.10 ^a^	0.80 ± 0.02 ^b^	427.81 ± 98.25 ^a^	378.85 ± 101.21 ^a^	0.77 ± 0.07 ^b^
38%	577.81 ± 203.36 ^a^	1.09 ± 0.45 ^a^	0.80 ± 0.02 ^b^	463.85 ± 165.34 ^a^	521.88 ± 136.29 ^a^	0.78 ± 0.04 ^b^
44%	691.09 ± 216.92 ^a^	0.94 ± 0.10 ^a^	0.76 ± 0.05 ^a^	533.85 ± 190.21 ^a^	508.67 ± 197.33 ^a^	0.70 ± 0.05 ^a^
50%	631.19 ± 192.72 ^a^	0.96 ± 0.06 ^a^	0.75 ± 0.03 ^a^	478.85 ± 153.32 ^a^	451.42 ± 153.52 ^a^	0.69 ± 0.05 ^a^
MIN	511.83	0.94	0.76	410.54	378.85	0.69
MAX	691.09	1.09	0.80	533.85	521.88	0.78
CV/%	12.45	6.12	2.90	10.39	13.07	6.32

MIN: minimum; MAX: maximum; CV: coefficient of variation. Data are the mean ± standard deviation of at least two replicates. Same letters in the same column of date signify not significant differences (*p* > 0.05).

## Data Availability

Data is contained within the article.

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
