# Peer review of "Impact of Oat Supplementation on the Structure, Digestibility, and Sensory Properties of Extruded Instant Rice"

_foods, 2024, doi:10.3390/foods13020217_

Round 1
Reviewer 1 Report
Comments and Suggestions for Authors
Title: Review Report for Manuscript - "Impact of Oat Supplementation on the Structure, Digestibility, and Sensory Properties of Extruded Instant Rice"
The manuscript explores the effects of oat supplementation on the structural, digestibility, and sensory properties of extruded instant rice. The study is well-conducted and addresses an interesting topic that holds significance in the food industry, considering the growing interest in the development of functional food products. The manuscript effectively combines analytical techniques to assess changes in the physical, chemical, and sensory characteristics of extruded instant rice upon oat supplementation. The incorporation of oats into extruded instant rice represents an innovative approach that aligns with the current trends in the food industry. The study adds valuable insights to the field by investigating the impact of this supplementation on multiple aspects of the final product.
Abstract: Add numeric observation in this section.
Introduction: Please elaborate this section by adding more information from latest scientific reports.
Justify the reason why authors used only specific percentage? Whether optimized using response surface methodology software or just the random selection it was?
The oat flour and broken rice flour were mixed in the proportions of 26%, 32%, 38%, 44%, and 50%.
Section 2.5.5. Sensory evaluation
Only 15 food professionals are not enough to judge the quality of specific products. The number must be higher………
Check the reference pattern
Reviewer 2 Report
Comments and Suggestions for Authors
The manuscript presents the results of the influence that addition of oat in various levels in instant extruded rice (IER) has on physicochemical, sensory and morphological changes of extruded products. The idea of incorporating oats in IER products is good one form a nutritional point of view in order to provide more dietary fibers in final food. Although the authors provide proper level of analytical characterization of all obtained products, the manuscript lacks some things
The introduction part is properly good, but section on material and methods requires extensive corrections of English, since it really hinders fluent reading in some parts, and also some additional explanation on methods have to be incorporated. Generally, the first part of the experiment lacks the extrusion of control sample, rice without any oat added and in the second part there is no clear explanation what was the sample in which optimal oat addition was executed. The set-up of the experiment requires to be better explained.
My comments are as follows:
Line 36: Give explanation of the abbreviation. This is the first time that is present in main text, excluding abstract. instant-extruded rice (IER).
Line 62: What was optimal oat addition?What were the optimization parameters that led the authors to choose IER-OA product and what kind of blend of rice and oats were optimal? Nothing is well defined and it should be much better explained. It is not present anywhere in the Results section what the optimal characteristics were. Just, I cannot see there was any optimization done.
Line 63: Oat instant rice or just instant rice?
Line 75: Delete word “it”.
Line 77: 80 mesh sieve. Delete a word “an”.
Lines 78-79: Please rephrase it. It is not clear which cereal was present at which level in the mixture.
Line 86: Please rephrase the first sentence in this line. It sounds like a recipe and not description of the experiment.
Line 89: Can you provide additional input parameters of the extrusion process - feed rate, screw speed, moisture content in the extruder barrel?
Line 89: Please provide number of knives of the cutter. Was it just one or more?
Lines 90-92: Provide the model and the name of the manufacturer of fluidized bed. What kind of oven was used? Was it continuous drier or some kind of other device? There are two zones mentioned. Provide the model and name of the manufacturer and give some more details on the oven.
Line 98: How the constant temperature was achieved? Does stirrer had a water bath for temperature control or the stirrer was put in some kind of climate chamber. Provide the model and name of the stirrer and centrifuge used.
Lines 106-114: The whole section needs to be rephrased and explained in a manner that does not feel like reading a method from a document. Use past tense and correct the English.
Line 117: What instructions? It was heated at which temperature and for how long?
Lines 120-121: Explained this in more details, do not just put the used parameters, please make a proper sentence.
Line 131: This is the first time of using this abbreviation in the text, excluding the abstract. Please use the term instant rice here and put IR abbreviation in brackets.
Lines 139-149: The English needs to be extensively corrected. Some parts are fine, but most parts feels like the authors give the method straight out of some handbook.
Line 177: English needs to be corrected.
Line 181: Breakdown Increased and decreased for what samples? Please specify.
Lines 188-189: What were the values of breakdown and setback and for which samples?
Line 206: What was the gelatinization temperature for this product?
Line 218: In each figure the legend needs correction. It should be stated before extrusion not "befor".
Lines 232-233: This construction and similar throughout he text should be be better defined. Specify with what oat addition there si increase and decrease, explain the figure in a better manner so that readers could follow easily by just reading text.
Line 274: An explanation of each abbreviation present in Figure 3 must be given so that potential readers can have all information available by just looking at the Figure.
Lines 302: The abbreviations RDS, SDS and RS need to be defined.
Comments on the Quality of English LanguageModerate editing of English language required for most of the manuscript, but extensive editing of English required for section Materials and methods.
Reviewer 3 Report
Comments and Suggestions for Authors
The experiment is interesting to readers. The findings of this study may be important information for the food development sector. In general, spelling, clarity, and linguistic accuracy should be reviewed. The specific comment was shown in the attachment file.
Specifics:
- The manuscript's flow should be rearranged.
- For the text, the SI unit should be utilized.
- The use of extruders in the food business should be addressed in the introductory section.
- Section 2.1: Please seek a commercially accessible supplier of oat instant rice (CAIR).
- Section 2.2: A reference to the manufacture of steamed oat flour should be provided.
- Is there a source for the oat flour/broken rice flour ratio utilized in this study? Why did the writers choose this percentage?
- How are the water absorption index (WAI), swelling power index (SPI), and water solubility index (WSI) calculated? The description clarified the preparatory steps and included no calculations or precise words to describe the measurement.
- The dietary fiber content calculation must be supplied.
- Because the sensory analysis was performed on humans, the ethical approval number for this investigation should be provided.
- Section 2.6 compares four distinct types of IR; consequently, the text explanation should include the name of each IR product. Furthermore, each product's preparation must be explained.
- Please provide the equation for computing the RDS, SDS, and RS values in Section 2.6.2.
- Line 140: The acetate buffer molarity was missing.
- Table 2: Pa×s should be used as the SI unit.
- Line 190: How did 26% oat addition to IR have anti-aging properties?
- Table 2: Each gelatinization characteristic or percentage of oat addition was compared in the table. According to the footnote, the comparison was based on gelatinization qualities.
- Figure 1A: Why did increasing the oat content to 50% enhance the gelatinization degree?
- Figure 1B did not have an explanation.
- Figures 1C and 1D show results that differ from Figure 1A.
- What is the dietary fiber food's standard value? Please incorporate it into the content. This information may be beneficial in comparison to the experiments.
- Figure 2: Consider categorizing the figure as A for fiber content, B for rehydration, and C for sensory ratings.
- What is the significance of the sensory score mentioned in Line 242 and Figure 2B?
- Table 3: How may the differences in each texture property be compared? If the value is not significantly different, the writers may put the footnote "ns=not significantly difference" above the texture attribute. Furthermore, the footnote to this Table should be clarified.
- Figure 3: Because of the different products utilized, please consider adjusting the chart style.
- The titles of Figures 4 and 5 should be moved to the bottom.
- Please verify the DOI and Font references.
Comments on the Quality of English LanguageEnglish editing is required.
Round 2
Reviewer 1 Report
Comments and Suggestions for Authors
Manuscript is thoroughly revised.
Reviewer 2 Report
Comments and Suggestions for Authors
The authors made effort to provide all corrections in the revised manuscript, which improved the overall paper.
The authors still need to address some issue and make corrections after which I would recommend the manuscript for publishing:
Lines 93-94: Liquid feed rate? What liquid? Does this refer to the water feed rate? Also, please provide this feed rate in an unit such as kg/h ,l/h etc. The feed rate in percentage is not well defined since it is not given 19% of what was liquid feed rate.
Lines 139 – 144: English still needs to be improved in first and the last sentence of the section.
Lines 155-156: The authors should provide the exact or at least the approximate temperature of rice cooker in which IER was heated for half an hour.
Comments on the Quality of English LanguageMinor editing of English language required
Reviewer 3 Report
Comments and Suggestions for Authors
Manuscript ID: foods-2790515
Type: Article
Title: Impact of Oat Supplementation on the Structure, Digestibility, and Sensory Properties of Extruded Instant Rice
Thank you for editing the manuscript in response to my concerns. Some issues are still to be addressed.
-Line 124: Where M is the sample mass (g) and M is the centrifuge tube mass (g), please indicate which is m or M.
- Line 134 now has Equation 4.
-Lines 135-136: where m represents the sample mass (g), mR represents the residue mass (g), mp represents the ash mass (g), mb represents the protein mass (g), and m represents the sample mass (g). Which m in the equation is related to ma and m?
-Line 155: Due to the immediate form of the IER sample, how should it be heated? Please elaborate.
-Lines 202, 203, and 204 now have Equations 5, 6, and 7.
-In section 2.5.3, formal English was used for scientific papers.
-The authors used capital letters to denote Figure 1, yet a tiny letter was used to show Figure 4. Please correct.
-Please correct the location of the image caption in image 4. 
-Figure 4 uses arrows to show the structural change in the IR sample.
-This document needs English revision.
Comments on the Quality of English Language
This document needs English revision.
